# Prognostic Significance of Perineural Invasion in Patients with Stage II/III Gastric Cancer Undergoing Radical Surgery

**DOI:** 10.3390/jpm12060962

**Published:** 2022-06-12

**Authors:** Yi-Fu Chen, Shan-Yu Wang, Puo-Hsien Le, Tsung-Hsing Chen, Chia-Jung Kuo, Chun-Jung Lin, Wen-Chi Chou, Ta-Sen Yeh, Jun-Te Hsu

**Affiliations:** 1Department of General Surgery, Chang Gung Memorial Hospital at Linkou, Chang Gung University College of Medicine, Taoyuan 33305, Taiwan; yifu061990@cgmh.org.tw (Y.-F.C.); m7026@cgmh.org.tw (S.-Y.W.); tsy471027@cgmh.org.tw (T.-S.Y.); 2Department of Gastroenterology, Chang Gung Memorial Hospital at Linkou, Chang Gung University College of Medicine, Taoyuan 33305, Taiwan; b9005031@cgmh.org.tw (P.-H.L.); q122583@cgmh.org.tw (T.-H.C.); m7011@cgmh.org.tw (C.-J.K.); ma1249@cgmh.org.tw (C.-J.L.); 3Department of Hematology-Oncology, Chang Gung Memorial Hospital at Linkou, Chang Gung University College of Medicine, Taoyuan 33305, Taiwan; f12986@cgmh.org.tw

**Keywords:** perineural invasion, gastric cancer, prognostic factor, survival

## Abstract

The prognostic significance of perineural invasion in patients with gastric cancer (GC) is controversial. This study aimed to determine the prognostic value of perineural invasion in patients with stage II/III GC undergoing radical surgery. A total of 1913 patients with stage II/III GC who underwent curative resection between 1994 and 2015 were recruited. Clinicopathological factors, tumor recurrence patterns, disease-free survival, and cancer-specific survival were compared in terms of perineural invasion. The prognostic factors of disease-free survival and cancer-specific survival were determined using univariate and multivariate analyses. Perineural invasion was found in 57.1% of the patients. Age of <65 years, female sex, large tumor size, upper tumor location, total gastrectomy, advanced tumor invasion depth and nodal involvement, greater metastatic to examined lymph node ratio, undifferentiated tumor, and presence of lymphatic or vascular invasion were significantly associated with perineural invasion. The patients with perineural invasion had higher locoregional/peritoneal recurrence rates than those without. Perineural invasion was independently associated with disease-free survival and cancer-specific survival. In conclusion, perineural invasion positivity is associated with aggressive tumor behaviors and higher locoregional/peritoneal recurrence rates in patients with stage II/III GC undergoing curative surgery. It is an independent unfavorable prognostic factor of disease recurrence and cancer-specific survival.

## 1. Introduction

Owing to improvements in the refrigeration and eradication of *Helicobacter pylori*, the incidence of gastric cancer (GC) has decreased steadily worldwide in recent decades [1,2,3]. However, GC remains the fourth leading cause of cancer-related death, despite advances in medicine and modern surgical technology [4]. The American Joint Committee on Cancer (AJCC)/Union for International Cancer Control (UICC) TNM staging system based on the depth of tumor invasion, number of regional lymph node (LN) metastases, and distant metastasis is commonly used to predict GC survival [5]. Nonetheless, patient prognosis still varies, even at the same stage. More prognostic factors, such as histological type, Lauren’s classification, and perineural invasion (PNI), are then used for survival analyses [6,7].

PNI is the process of the neoplastic invasion of the nerves, also called neurotropic carcinomatous and perineural spread. Some researchers define PNI as the presence of tumor cells in the nerve sheath, including the epineurium, perineurium, and endoneurium, or in the foci outside the nerve sheath with a 33% involvement of the nerve circumference [8]. PNI has been found to be a useful factor, analogous to vessel permeation, for assessing the malignant potential in a variety of malignancies, including head and neck, prostate, pancreatic, and colorectal cancers [8,9,10,11,12]. The UICC even added PNI as a parameter in the TNM classification for skin and colorectal cancers [13]. However, the role of PNI as a prognostic predictor in patients with GC remains controversial. Deng et al. conducted a systematic review and meta-analysis including 24 studies to explore the clinical significance of PNI in patients with GC and claimed its role as an outcome predictor with a weak point of significant heterogeneity among studies [7]. Similar results have been reported in another study [14]. Therefore, our study aimed to determine the prognostic impact of PNI on survival among a cohort of 1913 patients with stage II/III GC undergoing radical surgery in a tertiary medical center.

## 2. Materials and Methods

We retrospectively reviewed the medical records of 2752 patients with stage II/III gastric adenocarcinoma who underwent radical surgery between January 1994 and December 2015 at Chang Gung Memorial Hospital at Linkou (L-CGMH), Taiwan. Patients who had fewer than 16 LNs retrieved, surgical mortality (within 30 days after surgery), positive resection margins, and unknown lymphatic or vascular invasion status and who were lost to follow-up after surgery were excluded. Relevant data were retrieved from the institutional GC database of L-CGMH. Total or partial gastrectomy was performed on the basis of the tumor size, tumor location, and resection margin status. Frozen section examination for the resection margins was performed intraoperatively by pathologists. Fit patients underwent adjuvant chemotherapy 6–8 weeks after surgery. The eighth edition of the AJCC staging system was used for pathological tumor staging [5].

### 2.1. Clinical Information

Data on clinicopathological parameters, including age, sex, tumor size, tumor location, gastrectomy surgical procedures (total or partial), LN ratio (metastatic to examined LN ratio), histological type, lymphatic invasion, vascular invasion, PNI, TNM status, tumor staging, surgical complications, and adjuvant therapy administration were extracted from our institutional database.

### 2.2. Clinical Outcomes

Disease-free survival (DFS) was defined as the time from surgery to disease recurrence. Recurrences were categorized into three groups: locoregional recurrence, peritoneal seeding, and hematogenous spread. Locoregional recurrence was defined as local LN metastasis, extraluminal recurrence, recurrence within the gastric remnant stomach, or anastomotic recurrence following total gastrectomy. Peritoneal seeding was defined as the metastatic involvement of the peritoneum, such as intraperitoneal distribution or mesothelial implantation. Hematogenous spread/distant metastases included any visceral or distant nodal metastases (beyond locoregional nodes). Cancer-specific survival (CSS) was defined as the interval between the date of surgery and the date of death from GC. In addition to the impact of PNI on both DFS and CSS, the impact of PNI on the patterns of recurrence was also investigated. The median follow-up time was 38.24 months. The last follow-up date was 31 December 2020.

### 2.3. Statistical Analysis

Continuous variables were compared using Student’s *t*-test and categorical variables using Pearson’s chi-square test or Fisher’s exact test, as appropriate. Potentially relevant factors acquired from the univariate analysis (*p* < 0.05) were included in the multivariate analysis; both analyses were performed using Cox proportional hazard models, and a *p* value of <0.05 was considered statistically significant. Forest plots were used to compare the odds ratio (OR) of PNI in relation to the recurrence pattern. Patient survival was estimated using the Kaplan–Meier method, and the differences between subgroups were analyzed using the log-rank test. To analyze the prognostic factors of recurrence (tumor size and LN ratio), we performed a recursive partitioning analysis: a statistical methodology used to create a survival analysis tree and establish an optimal cut-off point for predicting recurrence [15]. All statistical analyses were conducted using the Statistical Package for the Social Sciences software for Windows (released 2011, version 20.0; IBM Corp., Armonk, NY, USA) and R software (R Core Team (2021), R: A language and environment for statistical computing. R Foundation for Statistical Computing, Vienna, Austria (https://www.R-project.org; accessed on 10 March 2022).

## 3. Results

A total of 1913 patients were included in our analysis. Table 1 demonstrates the clinicopathological characteristics of the patients with stage II/III GC. There were 1186 (62.0%) men and 727 (38.0%) women with a mean age of 63.7 years. The mean tumor size was 5.06 cm; the mean number of LNs retrieved was 36.5; and mean LN ratio was 0.23. The tumors were mostly located in the lower part of the stomach (58.5%), followed by the upper (20.0%) and middle (18.0%) parts of the stomach, entire stomach (2.9%), and other parts (0.6%). One thousand three hundred and five patients (68.2%) underwent partial gastrectomy. Differentiated histology was observed in 710 patients (37.1%). Lymphatic invasion, vascular invasion, and PNI were present in 1263 (66.0%), 336 (17.6%), and 1093 patients (57.1%), respectively. Moreover, 260 patients (13.6%) had surgical complications and 1401 (73.2%) received adjuvant chemotherapy. PNI positivity was identified in 57.1% of patients. Younger age (<65 years); female sex; larger tumor size; upper tumor location; number of LNs retrieved; total gastrectomy; advanced T status, N status, and stage; greater LN ratio; undifferentiated tumor; and presence of lymphatic or vascular invasion were significantly associated with PNI positivity. There was no difference in PNI positivity between surgical complications and the administration of adjuvant chemotherapy.

Tumor location, gastrectomy type, tumor size, T status, N status, stage, LN ratio, histological type, lymphatic invasion, vascular invasion, and PNI were found as the prognostic factors of DFS and CSS in the univariate analysis (Table 2 and Table 3). Meanwhile, tumor size, stage, LN ratio, and PNI were found as the independent predictors of DFS and CSS in the multivariate analysis (Table 4 and Table 5). Figure 1 depicts the Kaplan–Meier curve for DFS, indicating significantly lower 3-year and 5-year DFS rates in the patients with PNI than in those without (43.4% vs. 64.4% and 36.7% vs. 59.7%, respectively; *p* < 0.0001). The patients with PNI had worse 3-year and 5-year CSS rates than those without (50.7% vs. 69.6% and 39.4% vs. 61.8%, respectively; *p* < 0.0001), as shown in Figure 2.

Figure 3 shows the effect of PNI and the administration of adjuvant chemotherapy on the recurrence patterns in the patients with stage I/III GC who underwent radical surgery. A higher OR was identified for locoregional recurrence [OR = 1.623; 95% confidence interval (CI) = 1.300–2.025; *p* < 0.001] and peritoneal seeding (OR = 3.070; 95% CI = 2.354–4.004; *p* < 0.001) in the patients with PNI than in those without. The risk of peritoneal seeding in the patients with PNI who received adjuvant chemotherapy was significantly lower than that in those who did not receive adjuvant chemotherapy (OR = 2.631; 95% CI = 1.960–3.533 vs. OR = 5.558; 95% CI = 2.923–10.567; *p* = 0.038). Peritoneal seeding had a higher OR than locoregional recurrence in the patients with PNI who received adjuvant chemotherapy (*p* = 0.012).

## 4. Discussion

In our large-scale study, we investigated patients with stage II/III GC who underwent radical surgery. Our analyses indicated that PNI was present in 57.1% of the patients, in line with previously reported rates ranging from 31.7% to 65.0% [14]. Our study also showed that PNI was associated with younger age, female sex, upper stomach tumor location, total gastrectomy, and aggressive tumor behavior, including larger tumor size, undifferentiated histology, advanced stage, greater LN ratio, and the presence of lymphatic or vascular invasion. In the multivariate analysis, PNI was found as an independent prognostic factor of DFS and CSS.

Consistent with our study, previous studies have also revealed that larger tumor size, advanced AJCC stage, undifferentiated type, and presence of lymphatic or vascular invasion are associated with PNI positivity [14,16,17,18,19]. PNI is not only a signature feature of advanced GC but may also contribute to poor outcomes in the setting of relatively early disease. Our previous study found that PNI was an independent predictor of hematogenous metastasis in node-negative advanced GC [20]. Chen et al. also observed that PNI was an independent factor associated with early recurrence and poorer survival in patients with stage I–III GC after curative surgery [17]. Furthermore, two meta-analyses revealed that PNI independently affects DFS and overall survival in patients with resectable GC [7,14]. Therefore, PNI is not only an indicator of advanced GC, but may also exert its specific influence on the progression of GC regardless of the stage of cancer. Jiang et al. suggested a potential consideration of PNI in the future TNM staging system for GC [18]. Furthermore, some researchers have concluded that PNI might contribute to the stratification of individualized adjuvant therapy after surgery [12,21]. In support of their opinion, our analyses showed that patients with PNI who received adjuvant chemotherapy had a significantly lower risk of peritoneal seeding than those who did not receive chemotherapy (*p* = 0.038). Therefore, we suggest that advanced GC patients with PNI should undergo adjuvant chemotherapy to improve outcomes. Although more surgical and medical oncologists have adopted the concept that patients with PNI need more intensive adjuvant therapy, the role of PNI is still not incorporated into the AJCC/UICC system for predicting survival from GC.

Direct invasion of the vagus nerve is not easily identified preoperatively at the lesser curvature of the stomach, although patients with locally advanced GC have higher rates of PNI. PNI can still be noted, even in node-negative patients. Chen et al. suggested that skeletonization of the lesser curvature of the stomach during radical gastrectomy reduces local recurrence [22]. In addition, the application of an energy device with an ultrasonic scalpel makes it much easier to remove fat tissue and perform devascularization than the conventional technique of suture and ligation during surgery. The beneficial effects of skeletonization with lesser curvature on survival require further investigation in patients with GC undergoing radical surgery.

Our study observed a recurrence rate of 52.3% in patients with stage II/III GC who underwent radical surgery. Deng et al. reported that PNI had a negative impact on DFS, with a hazard ratio of 1.371 (95% CI = 1.230–1.527; *p* = 0.000) [7]. However, the association between recurrence patterns and PNI has rarely been investigated. Different disease statuses may influence the timing and pattern of recurrence [23]. Our study revealed that the risk of locoregional recurrence and peritoneal seeding significantly increased in the presence of PNI, and peritoneal seeding was even more prominent than locoregional recurrence under these conditions (*p* = 0.012). Further understanding of the mechanism by which PNI contributes to cancer cell spread-out may help to explain this phenomenon. In addition, more efforts are needed to develop therapeutic agents and approaches to reduce or inhibit PNI.

PNI has also been widely investigated and recognized as an indicator of aggressive tumor behavior in other malignancies [9,10,11,24]. For example, Knijn et al. reported that the incidence of PNI in patients with colorectal cancer was approximately 18.2% and increased with the severity of T and N status, poor differentiation, and the presence of lymphatic or vascular invasion. Furthermore, PNI has been associated with increased local recurrence rates and decreased survival rates in patients with colorectal cancer [11]. A similar pathological description has been observed in patients with pancreatic cancer [9]. Schorn et al. performed a systematic review of 121 studies indicating that patients with pancreatic cancer with PNI had significantly poor overall survival [9]. From the perspective of cancer biology, the interaction of cancer cells and peripheral nerves appears to be a vicious cycle: cancer cells cause nerve damage, and damaged nerves promote cancer cell spreading via the release of cytokines and chemokines [25,26,27]. PNI is a histological finding, and there are abundant interactions underlying this phenomenon. Recent basic and molecular studies also support the pathogenesis of PNI in a variety of malignancies [10,24,28]. Therefore, PNI is an indicator of aggressive behavior in different malignancies based on both clinicopathological observations and molecular findings.

Although our study results indicate the prognostic significance of PNI in patients with stage II/III GC undergoing radical surgery, this study has several limitations. First, inherent selection bias was unavoidable owing to the retrospective nature of the study. Second, the results were based on patients from a single medical center. Third, the patients received various chemotherapy regimens that might have influenced the survival outcomes. Fourth, we did not explore the underlying molecular mechanisms of PNI associated with poor prognosis and recurrence patterns.

## 5. Conclusions

PNI is associated with aggressive tumor behavior and locoregional/peritoneal recurrence and is also an independent poor prognostic factor of overall survival in patients with stage II/III GC undergoing curative surgery. A randomized control trial is needed to confirm whether more intensive adjuvant therapy is needed to improve prognosis in patients with PNI. In addition, the underlying pathophysiology and mechanisms of how PNI affects the recurrence patterns and prognosis of GC warrant further investigation.

## Figures and Tables

**Figure 1 jpm-12-00962-f001:**
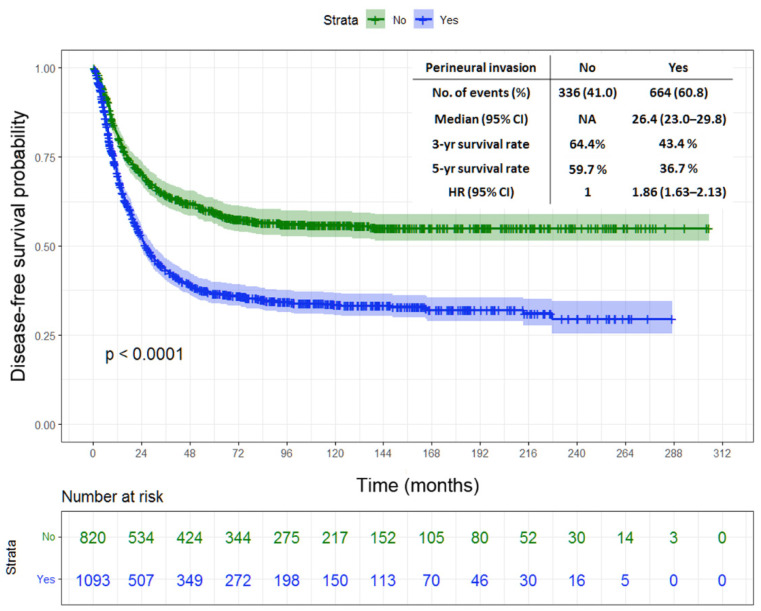
Kaplan–Meier survival curves of disease-free survival in patients with stage II/III gastric cancer undergoing radical surgery in terms of perineural invasion.

**Figure 2 jpm-12-00962-f002:**
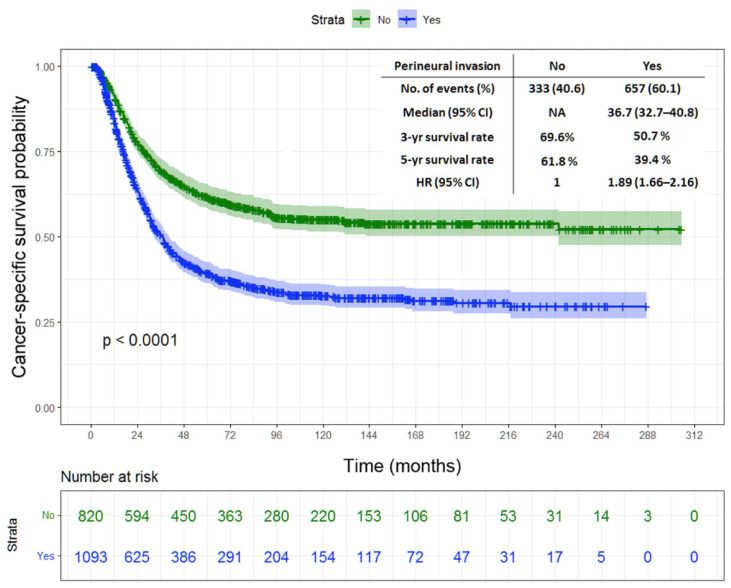
Kaplan–Meier survival curves of cancer-specific survival in patients with stage II/III gastric cancer undergoing radical surgery in terms of perineural invasion.

**Figure 3 jpm-12-00962-f003:**
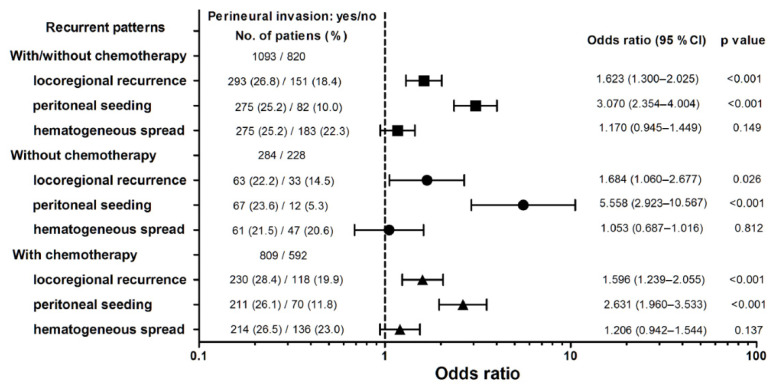
Forest plots of odds ratio with 95% confidence interval (CI) in recurrent patterns in terms of perineural invasion.

**Table 1 jpm-12-00962-t001:** Clinicopathological characteristics of the patients with stage II/III gastric cancer in terms of perineural invasion.

Parameters	Total	Perineural Invasion		*p* Value
		Negative	Positive	
Number	1913	820 (42.9)	1093 (57.1)	
Age (year), mean ± SD	63.68 ± 13.48	65.01 ± 12.75	62.68 ± 13.92	<0.001
≤65	935	374 (45.6)	561 (51.3)	0.013
>65	978	446 (54.4)	532 (48.7)	
Sex				0.040
Male	1186	530 (64.6)	656 (60.0)	
Female	727	290 (35.4)	437 (40.0)	
Tumor size (cm), mean ± SD	5.06 ± 2.84	4.82 ± 2.63	5.23 ± 2.98	0.002
Location				0.014
Upper	382	144 (17.6)	238 (21.8)	
Middle	344	155 (18.9)	189 (17.3)	
Lower	1119	501 (61.1)	618 (56.5)	
Entire	56	15 (1.8)	41 (3.8)	
Other	12	5 (0.6)	7 (0.6)	
Number of LNs retrieved, mean ± SD	36.50 ± 16.78	35.36 ± 16.38	37.35 ± 17.03	0.010
Type of gastrectomy				<0.001
Total	608	220 (26.8)	388 (35.5)	
Partial	1305	600 (73.2)	705 (64.5)	
T status				<0.001
T1	64	63 (7.7)	1 (0.1)	
T2	164	108 (13.2)	56 (5.1)	
T3	382	165 (20.1)	217 (19.9)	
T4a	1161	435 (53.0)	726 (66.4)	
T4b	142	49 (6.0)	93 (8.5)	
N status				<0.001
N0	332	195 (23.8)	137 (12.5)	
N1	308	175 (21.3)	133 (12.2)	
N2	459	219 (26.7)	240 (22.0)	
N3a	494	160 (19.5)	334 (30.6)	
N3b	320	71 (8.7)	249 (22.8)	
Stage (AJCC eighth edition)				<0.001
IIA	203	152 (18.5)	51 (4.7)	
IIB	357	214 (26.1)	143 (13.1)	
IIIA	548	241 (29.4)	307 (28.1)	
IIIB	454	135 (16.5)	319 (29.2)	
IIIC	351	78 (9.5)	273 (25.0)	
Histological type				<0.001
Differentiated	710	397 (48.4)	313 (28.6)	
Undifferentiated	1203	423 (51.6)	780 (71.4)	
Lymphatic invasion	1263	417 (50.9)	846 (77.4)	<0.001
Vascular invasion	336	66 (8.0)	270 (24.7)	<0.001
Surgical complication	260	103 (12.6)	157 (14.4)	0.255
Chemotherapy	1401	592 (72.2)	809 (74.0)	0.373

Data are presented as numbers (percentages), unless otherwise stated. AJCC—American Joint Committee on Cancer; SD—standard deviation; LN—lymph node.

**Table 2 jpm-12-00962-t002:** Univariate analysis of the prognostic factors of DFS in patients with stage II/III gastric cancer.

Factors	Median Survival (Month)	95% Confidence Interval	3-Year DFS (%)	5-Year DFS (%)	*p* Value
Age (year)					0.748
≤65 (*n* = 935)	46.7	32.7–60.8	52.6	46.9	
>65 (*n* = 978)	42.8	26.1–59.4	52.5	46.6	
Sex					0.417
Male (*n* = 1186)	41.0	30.8–51.2	51.4	45.6	
Female (*n* = 727)	50.3	28.1–72.5	54.3	48.6	
Location					<0.001
Upper (*n* = 382)	46.2	0.1–98.9	52.0	47.9	
Middle (*n* = 344)	148.4	NA	60.4	53.4	
Lower (*n* = 1119)	42.6	32.1–53.0	51.7	45.6	
Entire (*n* = 56)	18.1	12.6–23.6	31.1	26.4	
Other (*n* = 12)	10.5	3.5–17.5	13.3	13.3	
Type of gastrectomy					<0.001
Total (*n* = 608)	28.0	18.6–37.3	47.2	41.8	
Partial (*n* = 1305)	55.0	31.6–78.5	55.0	49.0	
Tumor size (cm)					<0.001
≤2.4 (*n* = 258)	NA		72.7	65.5	
24.1–3.9 (*n* = 485)	122.9	NA	59.5	52.6	
3.91–9.4 (*n* = 1006)	30.2	24.2–36.2	47.2	42.1	
>9.4 (*n* = 164)	14.2	8.8–19.6	32.0	27.7	
T status					<0.001
T1 (*n* = 64)	NA		67.3	61.9	
T2 (*n* = 164)	NA		74.7	65.4	
T3 (*n* = 382)	NA		58.6	53.5	
T4a (*n* = 1161)	34.2	26.7–41.7	49.2	43.4	
T4b (*n* = 142)	14.0	9.6–18.4	30.9	27.5	
N status					<0.001
N0 (*n* = 332)	NA		83.8	79.3	
N1 (*n* = 308)	NA		68.8	63.1	
N2 (*n* = 459)	139.9	NA	58.9	53.8	
N3a (*n* = 494)	21.8	18.9–24.7	34.1	27.2	
N3b (*n* = 320)	11.0	9.5–12.4	21.6	14.9	
Stage (AJCC eighth edition)					<0.001
IIA (*n* = 203)	NA		85.0	80.5	
IIB (*n* = 357)	NA		75.9	69.4	
IIIA (*n* = 548)	165.1	NA	60.1	55.0	
IIIB (*n* = 454)	21.9	18.9–24.8	34.0	28.1	
IIIC (*n* = 351)	11.5	10.1–12.8	20.6	14.2	
LN ratio					<0.001
≤0.05 (*n* = 566)	NA		79.4	75.2	
0.051–0.15 (*n* = 401)	NA		62.3	56.6	
0.151–0.37 (*n* = 479)	26.9	23.2–30.5	42.5	34.6	
0.371–0.53 (*n* = 217)	16.1	12.6–19.5	28.3	22.2	
>0.53 (*n* = 250)	9.9	8.4–11.4	14.3	8.8	
Histological type					<0.001
Differentiated (*n* = 710)	119.4	NA	58.9	53.6	
Undifferentiated (*n* = 1203)	31.9	25.0–38.9	48.7	42.6	
Lymphatic invasion					<0.001
No (*n* = 650)	NA	21.1–27.2	73.2	68.2	
Yes (*n* = 1263)	24.1	33.3–54.6	41.7	35.5	
Vascular invasion					<0.001
No (*n* = 1577)	59.7	15.0–104.4	55.9	50.0	
Yes (*n* = 336)	18.3	14.9–21.8	36.3	31.1	
Perineural invasion					<0.001
No (*n* = 820)	NA		64.4	59.7	
Yes (*n* = 1093)	26.4	23.0–29.8	43.4	36.7	
Chemotherapy					0.571
No (*n* = 512)	59.3	NA	56.0	49.7	
Yes (*n* = 1401)	41.4	31.4–51.3	51.5	45.9	

AJCC—American Joint Committee on Cancer; DFS—disease-free survival; LN ratio—metastatic to examined lymph node ratio; NA—not available.

**Table 3 jpm-12-00962-t003:** Univariate analysis of the prognostic factors of CSS in patients with stage II/III gastric cancer.

Factors	Median Survival (Month)	95% Confidence Interval	3-Year CSS (%)	5-Year CSS (%)	*p* Value
Age (year)					0.411
≤65 (*n* = 935)	62.4	46.9–78.0	60.3	50.5	
>65 (*n* = 978)	50.4	35.9–64.9	57.5	47.7	
Sex					0.374
Male (*n* = 1186)	52.2	39.9–64.4	57.6	47.9	
Female (*n* = 727)	62.7	45.1–80.2	61.1	51.1	
Location					<0.001
Upper (*n* = 382)	51.5	23.1–79.8	57.7	48.1	
Middle (*n* = 344)	100.4	-	66.5	55.7	
Lower (*n* = 1119)	55.5	44.1–66.9	58.4	48.9	
Entire (*n* = 56)	24.3	18.7–29.8	38.2	25.2	
Other (*n* = 12)	29.7	17.6–41.8	14.6	14.6	
Type of gastrectomy					<0.001
Total (*n* = 608)	38.4	28.9–47.9	51.5	43.4	
Partial (*n* = 1305)	66.7	51.9–81.4	62.3	51.7	
Tumor size (cm)					<0.001
≤2.4 (*n* = 258)	NA		79.6	67.8	
2.41–3.9 (*n* = 485)	126.3	NA	66.5	56.8	
3.91–9.4 (*n* = 1006)	41.6	34.6–48.6	53.7	43.8	
>9.4 (*n* = 164)	21.7	16.9–26.4	35.6	28.7	
T status					<0.001
T1 (*n* = 64)	241.9	0.1–486.8	75.3	68.0	
T2 (*n* = 164)	NA		80.2	70.3	
T3 (*n* = 382)	93.6	NA	67.4	56.2	
T4a (*n* = 1161)	43.7	35.6–51.7	54.9	45.1	
T4b (*n* = 142)	21.4	16.6–26.2	37.9	30.6	
N status					<0.001
N0 (*n* = 332)	NA		86.9	81.3	
N1 (*n* = 308)	NA		75.8	65.4	
N2 (*n* = 459)	118.9	58.6–179.3	64.3	55.7	
N3a (*n* = 494)	31.2	28.4–34.0	44.0	30.4	
N3b (*n* = 320)	18.3	15.6–21.1	27.9	18.1	
Stage (AJCC eighth edition)					
IIA (*n* = 203)	NA				<0.001
IIB (*n* = 357)	NA		86.7	82.4	
IIIA (*n* = 548)	104.1	NA	81.9	72.7	
IIIB (*n* = 454)	30.5	28.1–32.8	66.9	57.2	
IIIC (*n* = 351)	18.2	16.0–20.5	42.4	30.3	
LN ratio					<0.001
≤0.05 (*n* = 566)	NA		83.9	77.0	
0.051–0.15 (*n* = 401)	241.9	NA	68.4	58.6	
0.151–0.37 (*n* = 479)	38.1	32.4–43.9	52.2	38.3	
0.371–0.53 (*n* = 217)	24.4	19.7–29.1	35.8	25.4	
>0.53 (*n* = 250)	16.2	14.3–18.1	19.1	11.2	
Histological type					<0.001
Differentiated (*n* = 710)	100.7	31.3–170.1	65.9	56.0	
Undifferentiated (*n* = 1203)	44.0	36.6–51.5	54.8	45.1	
Lymphatic invasion					<0.001
No (*n* = 650)	NA		78.7	70.7	
Yes (*n* = 1263)	33.8	30.7–37.0	48.6	37.9	
Vascular invasion					<0.001
No (*n* = 1577)	72.2	48.6–95.8	62.4	52.5	
Yes (*n* = 336)	26.7	21.4–32.1	42.4	32.9	
Perineural invasion					<0.001
No (*n* = 820)	NA		69.6	61.8	
Yes (*n* = 1093)	36.7	32.7–40.8	50.7	39.4	
Chemotherapy					0.862
No (*n* = 512)	62.6	39.4–85.7	60.6	50.8	
Yes (*n* = 1401)	55.5	44.0–66.9	58.5	48.7	

AJCC—American Joint Committee on Cancer; CSS—cancer-specific survival; LN ratio—metastatic to examined lymph node ratio; NA—not available.

**Table 4 jpm-12-00962-t004:** Multivariate analysis of the prognostic factors of disease-free survival in patients with stage II/III gastric cancer.

Factors	Hazard Ratio	95% Confidence Interval		*p* Value
		Lower	Upper	
Type of gastrectomy				
Total/partial	1.149	1.000	1.319	0.050
Tumor size (cm)				
2.41–3.9/≤2.4	1.167	0.913	1.491	0.217
3.91–9.4/≤2.4	1.397	1.113	1.754	0.004
>9.4/≤2.4	1.581	1.177	2.124	0.002
Stage (AJCC eighth edition)				
IIB/IIA	1.598	1.108	2.306	0.012
IIIA/IIA	1.766	1.232	2.531	0.002
IIIB/IIA	1.998	1.344	2.971	<0.001
IIIC/IIA	2.283	1.493	3.492	<0.001
LN ratio				
0.051–0.15/≤0.05	1.595	1.239	2.053	<0.001
0.151–0.37/≤0.05	2.364	1.817	3.075	<0.001
0.371–0.53/≤0.05	3.116	2.270	4.276	<0.001
>0.53/≤0.05	4.570	3.288	6.354	<0.001
Histological type				
Differentiated/undifferentiated	1.029	0.896	1.182	0.685
Vascular invasion				
Yes/no	1.074	0.918	1.256	0.374
Lymphatic invasion				
Yes/no	1.161	0.963	1.400	0.118
Perineural invasion				
Yes/no	1.227	1.063	1.415	0.005

AJCC—American Joint Committee on Cancer; LN ratio—metastatic to examined lymph node ratio.

**Table 5 jpm-12-00962-t005:** Multivariate analysis of the prognostic factors of cancer-specific survival in patients with stage II/III gastric cancer.

Factors	Hazard Ratio	95% Confidence Interval		*p* Value
		Lower	Upper	
Type of gastrectomy				
Total/partial	1.149	0.999	1.320	0.051
Tumor size (cm)				
2.41–3.9/≤2.4	1.189	0.927	1.525	0.173
3.91–9.4/2.4	1.439	1.142	1.813	0.002
>9.4/≤2.4	1.692	1.254	2.282	<0.001
Stage (AJCC eighth edition)				
IIB/IIA	1.558	1.075	2.259	0.019
IIIA/IIA	1.796	1.249	2.583	0.002
IIIB/IIA	2.082	1.396	3.106	<0.001
IIIC/IIA	2.357	1.535	3.619	<0.001
LN ratio				
0.051–0.15/≤0.05	1.547	1.201	1.993	<0.001
0.151–0.37/≤0.05	2.249	1.728	2.928	<0.001
0.371–0.53/≤0.05	2.943	2.145	4.039	<0.001
>0.53/≤0.05	4.331	3.114	6.024	<0.001
Histological type				
Differentiated/undifferentiated	1.058	0.921	1.215	0.427
Vascular invasion				
Yes/no	1.062	0.906	1.243	0.459
Lymphatic invasion				
Yes/no	1.168	0.969	1.408	0.104
Perineural invasion				
Yes/no	1.231	1.066	1.421	0.005

AJCC—American Joint Committee on Cancer; LN ratio—metastatic to examined lymph node ratio.

## Data Availability

Not applicable.

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
