# Peer review of "Prognostic Significance of Perineural Invasion in Patients with Stage II/III Gastric Cancer Undergoing Radical Surgery"

_jpm, 2022, doi:10.3390/jpm12060962_

Round 1

Reviewer 1 Report

In this manuscript, the authors investigated 1913 patients with stage II/III GC (Gastric Cancer) who underwent radical surgery. Their results showed that PNI (perineural invasion) was associated with aggressive tumor behavior and locoregional/peritoneal recurrence. And they also demonstrated that PNI is also an independent poor prognostic factor of overall survival in patients. This manuscript might be a good restrospective summary of previous GC patients cases. In the manuscript, the author have discussed that PNI is an independent poor prognostic factor. To further improve the manuscript, It will be better for authors to discuss the methods for detecting PNI, how to reduce or inhibit PNI and what’s the recommend therapies if PNI is occurred.

Author Response

Comments

In this manuscript, the authors investigated 1913 patients with stage II/III GC (Gastric Cancer) who underwent radical surgery. Their results showed that PNI (perineural invasion) was associated with aggressive tumor behavior and locoregional/peritoneal recurrence. And they also demonstrated that PNI is also an independent poor prognostic factor of overall survival in patients. This manuscript might be a good retrospective summary of previous GC cases. In the manuscript, the authors have discussed that PNI is an independent poor prognostic factor. To further improve the manuscript, it will be better for authors to discuss the methods for detecting PNI, how to reduce or inhibit PNI and what’s the recommend therapies if PNI is occurred. 

Responses: We thank the reviewer very much for the valuable comments regarding our work. PNI is diagnosed microscopically from the surgical specimen by the pathologist in this study. It is difficult to identify PNI preoperatively. Besides, there is still no evidence or methods addressing how to reduce or inhibit PNI in the literature. Accordingly, we added a sentence to emphasize the unmet need in the 4th paragraph of discussion.

Further understanding of the mechanism by which PNI contributes to cancer cell spread-out may help to explain this phenomenon. In addition, more efforts are needed to develop therapeutic agents or approaches to reduce or inhibit PNI.  

Our results showed that patients with PNI who received adjuvant chemotherapy had a significantly lower risk of peritoneal seeding than those who did not receive chemotherapy (p = 0.038). Accordingly, we added one sentence to recommend therapy if PNI is occurred in the 2nd paragraph of discussion.

Furthermore, some researchers have concluded that PNI might contribute to the stratification of individualized adjuvant therapy after surgery [12,21]. In support of their opinion, our analyses showed that patients with PNI who received adjuvant chemotherapy had a significantly lower risk of peritoneal seeding than those who did not receive chemotherapy (p = 0.038). Therefore, we suggest that advanced GC patients with PNI should undergo adjuvant chemotherapy to improve outcomes. Although more surgical and medical oncologists have adopted the concept that patients with PNI need more intensive adjuvant therapy, the role of PNI is still not incorporated into the AJCC/UICC system for predicting survival from GC.

Reviewer 2 Report

Thanks for the opportunity to review the manuscript "Prognostic Significance of Perineural Invasion in Patoents with Stage II/III Gastric Cancer Undergoing Radical Surgery" by Chen et al. It is a well written and scientific prepared article about evaluation of the aggressivity of gastric tumors defined by the perineural invasion. The authors thus manage a successful re-evaluation of regularly collected parameters. I have no major criticisms to note, only a presentation of the results without the very space-consuming tables would be desirable. Perhaps a better clarity can be achieved by reducing the collected parameters? Overall, I support the publication of the presented manuscript.  

Author Response

Comments:

Thanks for the opportunity to review the manuscript "Prognostic Significance of Perineural Invasion in Patients with Stage II/III Gastric Cancer Undergoing Radical Surgery" by Chen et al. It is a well written and scientific prepared article about evaluation of the aggressivity of gastric tumors defined by the perineural invasion. The authors thus manage a successful re-evaluation of regularly collected parameters. I have no major criticisms to note, only a presentation of the results without the very space-consuming tables would be desirable. Perhaps a better clarity can be achieved by reducing the collected parameters? Overall, I support the publication of the presented manuscript.

Response: We thank the reviewer greatly for comments that our manuscript is well-written and scientifically prepared. The parameters included in the tables are important for presentation and GC prognosis. We have revised the table 1 to reduce space as the reviewer’s suggestion.
